# CRISPR/Cas13a-based supersensitive circulating tumor DNA assay for detecting *EGFR* mutations in plasma
Li Wang [1,6], Xiaosha Wen [2,3,6], Yang Yang [2,3], Zheng Hu [4], Jing Jiang [4], Lili Duan [4], Xiaofen Liao [4], Yan He [2,3], Yaru Liu [2,3], Jing Wang [2,3], Zhikun Liang [5], Xiaoya Zhu [5], Quan Liu [2,3] ✉, Tiancai Liu [1] ✉ & Dixian Luo [2,3] ✉

Despite recent technological advancements in cell tumor DNA (ctDNA) mutation detection, challenges persist in identifying low-frequency mutations due to inadequate sensitivity and coverage of current procedures. Herein, we introduce a super-sensitivity and specificity technique for detecting ctDNA mutations, named HiCASE. The method utilizes PCR-based CRISPR, coupled with the restriction enzyme. In this work, HiCASE focuses on testing a series of *EGFR* mutations to provide enhanced detection technology for non–small cell lung cancer (NSCLC), enabling a detection sensitivity of 0.01% with 40 ng cell free DNA standard. When applied to a panel of 140 plasma samples from 120 NSCLC patients, HiCASE exhibits 88.1% clinical sensitivity and 100% specificity with 40 μL of plasma, higher than ddPCR and Super-ARMS assay. In addition, HiCASE can also clearly distinguish T790M/C797S mutations in different positions at a 1% variant allele frequency, offering valuable guidance for drug utilization. Indeed, the established HiCASE assay shows potential for clinical applications.

Circulating tumor DNA (ctDNA) as a liquid biopsy, released from tumor cells, presents a non-invasive and promising avenue for identifying genetic alterations[1–4]. For example, *EGFR* is the most common oncogenic driver gene in non–small cell lung cancer (NSCLC), where exon 19 deletion mutation (19-del) and exon 21 L858R account for 90%[5,6]. Notably, EGFR-tyrosine kinase inhibitors have a better effect on *EGFR*-related mutations. By employing ctDNA, mutations can be identified in blood samples, offering a viable alternative to conventional tissue biopsies. Nevertheless, in the early and mid stages of lung cancer, the variable DNA shedding from tumor is relatively low and surrounded by the wild-type DNA shed from normal cells. Moreover, ctDNA displays highly fragmented characteristics and a short half-life, further complicating detection with conventional technologies[7,8].

Some highly sensitive approaches have been developed to detect low-frequency mutations in clinical settings[9–11]. For example, PCR-based methods mainly include the amplification refractory mutation system (ARMS) and digital PCR. The ARMS-based allele-specific assay has been utilized for detecting *EGFR*-sensitizing mutations in tissue samples because of its simple operation and time efficiency. However, considering limited sensitivity (0.2–1%), ARMS is not sufficient to detect low-frequency

mutations in plasma samples, and the method is susceptible to generating false-positive signals because of the PCR primer's specificity. Digital PCR exhibits excellent sensitivity and specificity for rare variant alleles but requires high-cost equipment. Sequencing-based methods, such as next-generation sequencing (NGS), can provide information for multiple genes, while it poses difficulty in detecting allelic mutants present below 0.1% with routine sequencing depth. Cancer personalized profiling by deep sequencing is a highly sensitive NGS-based technology developed at Stanford University for ctDNA detection[12]. It's gradually being applied in various cancer studies for diagnosis, profiling, and treatment response tracking. However, PCR-related methods may be more economical and faster for detecting several target genes. The application of restriction endonucleases (REs) for identifying specific nucleic acids is also a workable technical approach that has led to the development of methods for detecting ctDNA mutations[13,14]. However, although REs can specifically digest wild-type fragments and be combined with PCR amplification, their application is hampered by low sensitivity.

Recently, the CRISPR/Cas (Clustered Regularly Interspaced Short Palindromic Repeats/CRISPR-associated protein) system has become recognized for its high efficiency in sgRNA (Single guide RNA)-mediated

[1]Key Laboratory of Antibody Engineering of Guangdong Higher Education Institutes, School of Laboratory Medicine and Biotechnology, Southern Medical University, Guangzhou 510515, PR China. [2]Department of Laboratory Medicine, Huazhong University of Science and Technology Union Shenzhen Hospital (Nanshan Hospital), Shenzhen 518052, PR China. [3]Shenzhen University Medical School, Shenzhen 518060, PR China. [4]Translational Medicine Institute, the First People's Hospital of Chenzhou Affiliated to University of South China, Chenzhou 423000, PR China. [5]Research Institute, DAAN Gene Co., Ltd., Guangzhou 510665, PR China. [6]These authors contributed equally: Li Wang, Xiaosha Wen. ✉e-mail: liu_quan2020@163.com; liutc@smu.edu.cn; luodixian_2@163.com

cleavage of a target nucleic acid. Especially, the Cas12 and Cas13 proteins are widely used to establish molecular diagnostic platforms[15–18]. Upon recognizing the target sequence, the Cas/sgRNA binary complex becomes activated, leading to the nonspecific cleavage of other single-stranded (ss) DNA or ssRNA probes. According to the literature, SHERLOCK technology can discern single-nucleotide variant levels ranging from 1 to 0.1%[19], but it does not show obvious advantages compared with PCR-based detection methods. Although introducing two or three mismatches between crRNA and the target sequence increases the specificity of mutant-crRNA, there are still fluorescent signals toward wild-type fragments. Simultaneously, this leads to a visible decrease in the mutant-crRNA's ability to target sites. Scientists attend to engineer crRNA structure and Cas proteins enhancing the nuclease specificity and activity[20–22]. This is a trustworthy means of mitigating off-target effects in vivo but has a reduced impact on in vitro diagnostic applications.

To increase the sensitivity and specificity of the Cas13/crRNA ctDNA mutation assay, we introduced REs to selectively digest wild-type DNA fragments, coupling the high sensitivity of PCR-Cas13a for detection, termed HiCASE (High-sensitivity PCR-Cas13a with Specific restriction Enzyme detection). Indeed, the Cas13a enzyme displayed a remarkable trans-cleavage efficiency and only provided a few mutant nucleic acids, allowing for the detection of several copies of the mutation. To evaluate the clinical performance of the HiCASE assay, 140 plasma samples obtained from NSCLC patients and healthy donors were included to detect *EGFR* mutations using three different approaches, namely HiCASE, ddPCR, and Super-ARMS assays. The results revealed that our method exhibited high sensitivity and specificity compared with digital PCR and Super-ARMS assay.

## Results

### Highly sensitive and specific detection of *EGFR* point mutations by HiCASE

We introduced a ctDNA mutation detection assay, named HiCASE, which involves PCR pre-amplification, digestion with an RE, and subsequent detection utilizing the Cas13a/crRNA system (Fig. 1). In our study, we mainly focused on *EGFR* mutations, selecting L858R, 19dels, T790M, and C797S sites—known for their higher mutation frequencies in NSCLC patients—as the targets of the HiCASE assay. To achieve discrimination between the L858R wild-type and mutant-type sequences, eight specific-mutant crRNAs were designed to screen out optimal targeting efficiency (Supplementary Fig. 1a). We showed that crRNA3 exhibited better efficiency in detecting the L858R target based on the fluorescence signal and

was further verified on urea gel electrophoresis (Fig. 2a). In general, the crRNA preparation process involves obtaining the DNA template for transcription, conducting in vitro transcription, and performing RNA purification, which is laborious and time-consuming. Here, our data exhibited the DNA template had comparable target efficiency to the crRNA (Fig. 2b). While an elevated $MgCl_2$ concentration enhanced the cleavage efficiency of Cas13a/crRNA, it concurrently decreased the specificity of Cas13a/crRNA, leading to an increase in nonspecific fluorescent signals. We determined that 9 mM $MgCl_2$ showed better compatibility (Supplementary Fig. 1b). To further reduce the interference of wild-type fragments on detection, the MscI enzyme was utilized to selectively digest wild type (Supplementary Fig. 1c–d). We showed that the MscI enzyme had a better specificity for digesting wild type and effectively diminished the fluorescent signal. We mixed the plasmids of the mutant type and wild type at a series of ratios. The result verified that the detection sensitivity was obviously increased through the incorporation of an RE into the Cas13a/crRNA detection (Fig. 2c). Furthermore, digesting wild fragments with RE can reduce their interference in fluorescence detection, resulting in a noticeable improvement in fluorescent values for detecting mutant fragments. Different variant allele frequencies (VAFs) of L858R cell free DNA (cfDNA) standards were used to evaluate the detection sensitivity of HiCASE (Fig. 2d). The results showed that there was no cross-reactivity with the wild-type fragments, and the VAF as low as 0.02% was achieved using 20 ng cfDNA input, which included two copies of mutant fragments quantified by ddPCR. HiCASE can also identify a mutation at a sensitivity of 0.01% when provided with 40 ng of cfDNA as input (Supplementary Fig. 1e). In addition, the fluorescent value showed a linear dependence on different VAFs (Fig. 2e), and the limit of detection (LOD) was determined by testing the wild type cfDNA repeatedly 10 times (Supplementary Fig. 1f).

To address the absence of RE sites in the target sequence, we designed mutagenic primers of T790M and incorporated the substituted base into the PCR product. Due to Taq DNA polymerase lacking 3' to 5' exonuclease activity, it cannot identify a mismatch between the primer and the template. After several cycles of PCR, most of the products will harbor the mismatch base (Supplementary Fig. 2). The Sanger sequencing result showed that the desired base substitution (A > C) was successfully introduced into the target sequence (Fig. 2f). The PCR products were digested with the SsiI enzyme, as indicated in Fig. 2g. We found that wild-type DNA fragments were digested, whereas mutant-type fragments remained undigested. Subsequently, specific-mutant crRNA6 was screened out as the optimal targeting efficiency (Supplementary Fig. 3a). The HiCASE result revealed that the VAF of 0.05% T790M was detectable, and it was a clear linear correlation between

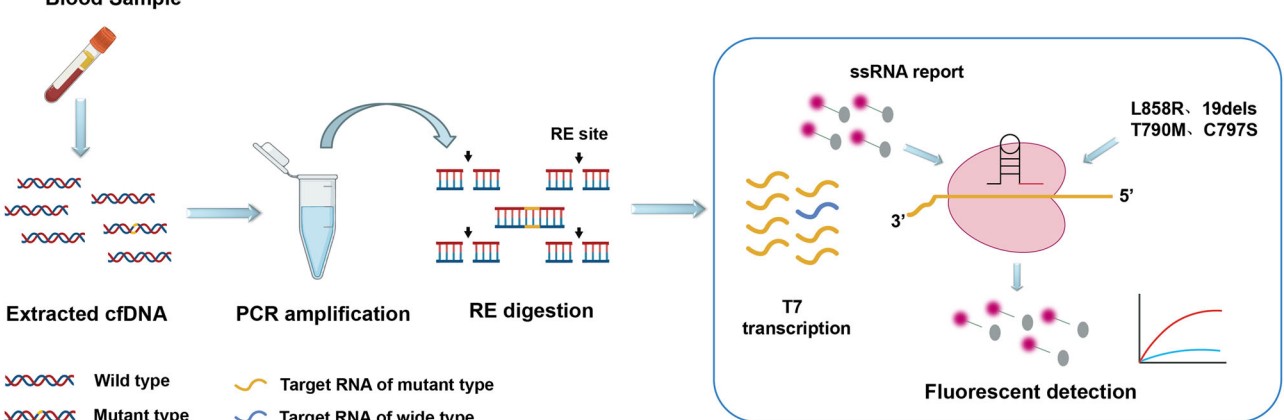

**Fig. 1 | Schematic illustration of the HiCASE assay for the detection of cfDNA sample.** The HiCASE assay relies on the CRISPR/Cas13a system coupled with PCR amplification and RE digestion. It consists of three main steps. First, cfDNA extracted from the plasma sample is amplified by PCR. Then, the PCR products are selectively digested by RE when the target sequence carries the RE site. Finally, the digested products are detected by Cas13a/crRNA. In this mixture, DNA products are transcribed into target RNA through the T7 promoter, and then the Cas13a/crRNA complex recognizes the target RNA via the spacer sequence and cleaves it. As a result, Cas13a is triggered to nonspecifically cleave the ssRNA reporters, detectable through a fluorescent instrument.

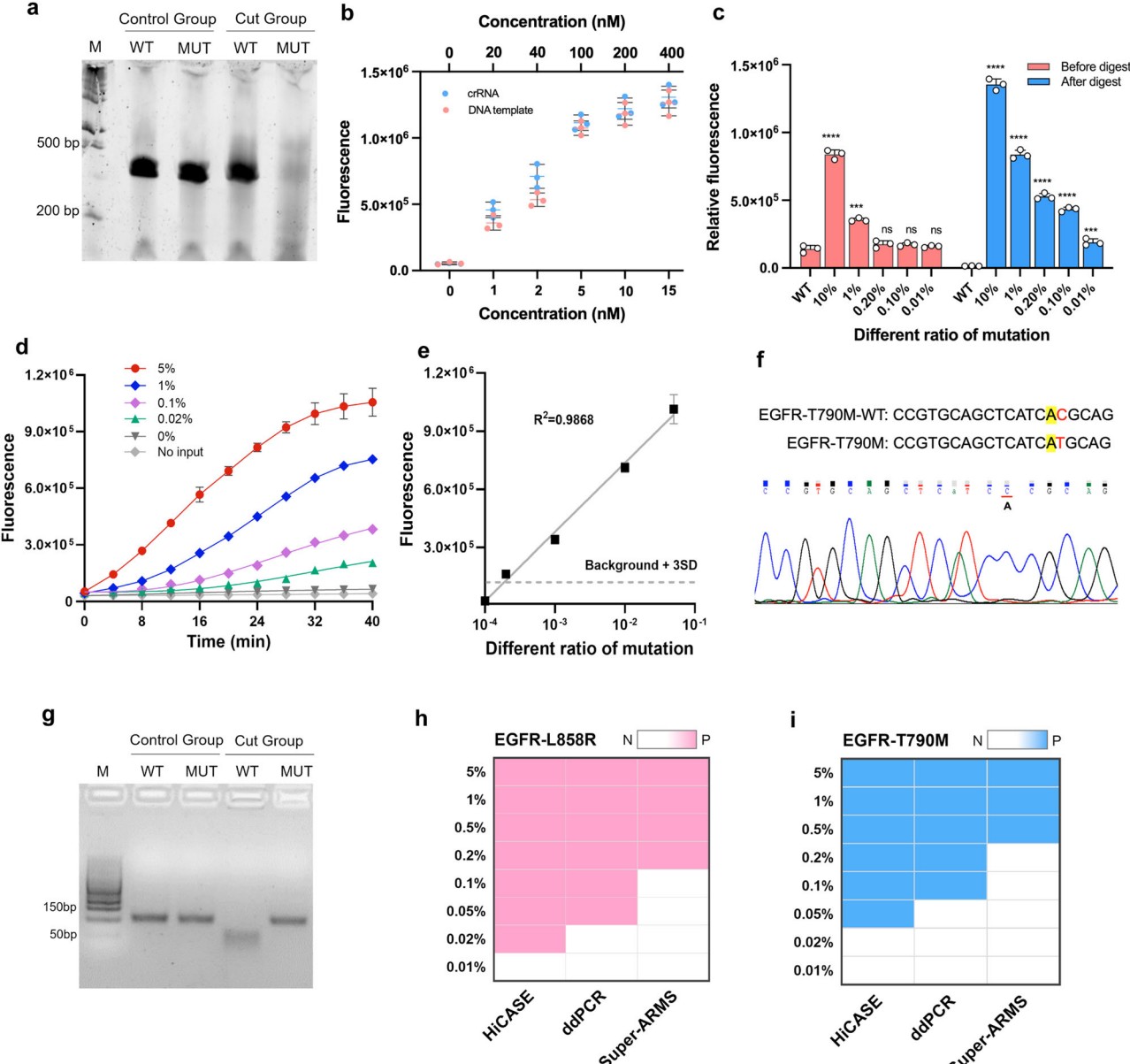

**Fig. 2 | Detecting *EGFR* L858R and T790M mutations using the HiCASE assay.**
**a** The RNA denaturing Urea PAGE for analysis of *EGFR* L858R wild type and mutant type RNA target fragments before and after cleaved by Cas13a/crRNA complex. WT, wide type; MUT, mutant type; M, 200 bp marker of RNA.
**b** Comparison of the target efficiency between DNA template and crRNA in different concentrations. **c** The detection of different VAFs of *EGFR* L858R plasmids using two approaches. **d** The real-time fluorescent curves of L858R using HiCASE assay for detecting different VAFs with cfDNA standards. **e** The standard curve of L858R mutation detection. **f** The site-directed mutagenesis of *EGFR* T790M and the result of Sanger sequencing. Yellow indicated the base to be mutated and red for mutated base of wild type. The red line indicated the base after mutated. **g** The electropherogram of PCR products of T790M before and after being digested by the SsiI enzyme. **h**, **i** Comparing the detection sensitivity of ctDNA in L858R and T790M mutations using HiCASE, ddPCR, and Super-ARMS assays, respectively. The data was analyzed using two-tailed Student's *t* test. *n* = 3 independent experiments, The error bars indicate standard deviation; *P < 0.05, **P < 0.01, ***P < 0.001, and ****P < 0.0001; ns, not significant.

the fluorescent values with various VAFs (Supplementary Fig. 3b–c). The LOD was confirmed by testing the wild type cfDNA repeatedly 10 times (Supplementary Fig. 3d). Next, we evaluated the sensitivity in detecting point mutations by comparing HiCASE, ddPCR, and Super-ARMS assays with cfDNA standards (Fig. 2h–i). The results showed that HiCASE exhibited superior sensitivity in identifying L858R and T790M mutations, indicating its considerable potential for the detection of rare VAFs.

**Adapting the HiCASE assay for detecting *EGFR* 19dels mutations**
*EGFR* 19del contains a series of deletion sites. In theory, our HiCASE assay can cover multiple deletion sites by adding deletion-specific crRNAs into the

Cas13a/crRNA system. Herein, we chose three deletion subtypes (delE746_A750, delL747_P753insS, delL747_T751) that are the most frequently occurring in *EGFR* 19dels mutation. Four crRNAs (del15-1, del15-2, del18, del15-3) were screened out to specifically target the deletion sites (Fig. 3a). For each deletion site, eight crRNAs were designed to identify the optimal one with high target efficiency (Fig. 3b). Based on the fluorescent values, we found that del15-1-crRNA2, del15-2-crRNA2, del18-crRNA1, and del15-3-crRNA6 exhibited superior sensitivity and specificity toward the deletion sites. The result presented in Fig. 3c validated that the MseI enzyme can selectively digest the wild-type sequence. Following digestion by MseI, the dels15-1 mutation had a notable enhancement in the detection

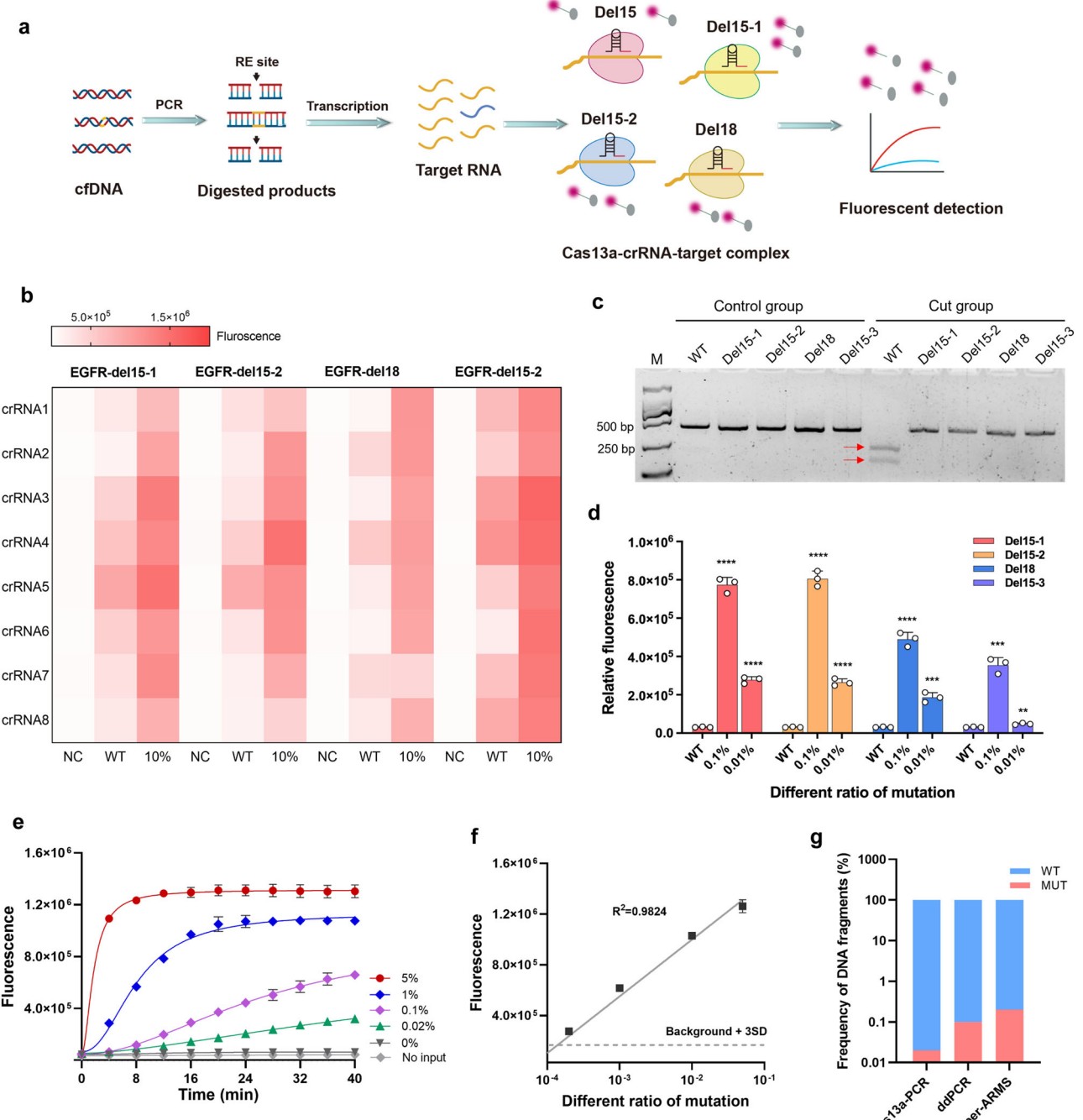

**Fig. 3 | Detecting *EGFR* 19dels mutations using the HiCASE assay. a** The workflow for detecting *EGFR* 19dels by HiCASE assay. cfDNA was subjected to PCR amplification and RE digestion. In the third step, four crRNAs targeting different deletion mutations were added to the Cas13a/crRNA reaction. del15-1, (E746_A750del; del-2235-2249); del15-2, (E746_A750del; del-2236-2250); del18, (delL747_P753insS; del-2240-2257); del15-3 (delL747_T751; del-2240-2254). **b** Screening out the optimal crRNA for each deletion mutation from a pool of eight crRNAs. 10% VAF was mixed with 20 ng wild type and 2 ng mutant type templates. **c** The electropherogram showed the specificity of the MseI enzyme for digesting wild type and 19dels target fragments. M, 2000 bp marker of DNA. The red arrow represented cleaved bands. **d** The detection for different VAFs (0.1%, 0.01%) mixed by wild type and mutant type plasmids using HiCASE assay. **e** The real-time fluorescent curves of del15-1 mutation using HiCASE for detecting different VAFs with cfDNA standards. **f** The standard curve of del15-1 mutation indicated the correlation of fluorescent value with different VAFs by HiCASE. **g** Comparing the detection sensitivity of del15-1 mutation with HiCASE, ddPCR, and Super-ARMS assays, respectively. The data was analyzed using two-tailed Student's *t* test. *n* = 3 independent experiments. error bars represent the mean ± S. The error bars indicate standard deviation; *$P < 0.05$, **$P < 0.01$, ***$P < 0.001$, and ****$P < 0.0001$.

sensitivity. (Supplementary Fig. 4). By employing plasmids as input templates and then utilizing HiCASE for detection, mutant alleles at a level of 0.01% were also detected in the del15, del15-1, del18, and del15-3 deletion sites (Fig. 3d). Next, different VAFs of cfDNA standards were used to determine the reliability of HiCASE in detecting deletion mutation. The

HiCASE assay was able to detect del15-1 mutation as low as 0.02% and showed a better linear correlation between fluorescent values and different VAFs of the del15-1 mutation (Fig. 3e, f). The detection limit was obtained by repeatedly testing ten times of wild type cfDNA standard (Supplementary Fig. 5). The HiCASE assay also showed a higher sensitivity in detecting

del15-1 mutation compared with ddPCR and super-ARMS assays (Fig. 3g). In summary, the HiCASE assay holds promise for application in the analysis of clinical plasma samples.

## Detection for cfDNA extracted from NSCLC plasma samples with different assays

To test the clinical sensitivity and specificity of the HiCASE assay, 140 plasma samples of NSCLC patients and healthy donors were collected as shown in Fig. 4a. In this study, the plasma samples from the NSCLC patients were paired with tissue samples, and the analysis of *EGFR* mutation in tumor tissue was considered the gold standard. Among the 120 NSCLC patients, 35 cases had *EGFR* mutations, including 18 with the L858R mutation and 17 with the 19dels mutation. In the subset of 20 patients treated with TKIs, 7 cases exhibited the T790M mutation. The incidence rates for L858R, 19dels, and T790M were 15% (18/120), 14.2% (17/120), and 35% (7/20), respectively. The concentrations of cfDNA extracted from NSCLC plasmas ranged from 9 ng/mL to 612 ng/mL, with a median concentration of 62 ng/mL (Supplementary Fig. 6a). The HiCASE assay was employed to identify *EGFR* mutations of L858R and 19dels (Fig. 4b). We found that the negative and positive samples were clearly distinguishable based on their fluorescent values. Analysis of ctDNA revealed L858R in 13.3% (16/120), 19dels in 12.5% (15/120), and T790M in 30% (6/20) of plasma samples. We further compared the clinical performance of HiCASE with ddPCR and Super-ARMS assays for *EGFR* T790M mutations (Fig. 4c). The number of positive samples determined by HiCASE, ddPCR, and Super-ARMS assays was 6, 3, and 3, respectively. As for the L858R mutation, HiCASE, ddPCR, and Super-ARMS assays demonstrated sensitivity of 88.9 (16/18), 66.7 (12/18), and 55.6% (10/18), with specificity of 100 (102/102), 99.0 (101/102), and 98.0% (100/102), respectively. For the 19dels mutation, the sensitivity and specificity of the three assays were 88.2 (15/17) and 100% (103/103), 64.7 (11/17) and 100% (103/103), 58.8 (10/17) and 98.1% (101/103), respectively (Fig. 4d, e and Supplementary Data 1). The accuracy of *EGFR* mutations detected by HiCASE, ddPCR, and Super-ARMS assays was 98.1 (255/260), 93.5 (243/260), and 91.2% (237/260), respectively (Supplementary Data 2). Among 23 samples identified with L858R or 19dels mutations via ddPCR, the VAFs ranged from 0.13% to 15.4% (Supplementary Fig. 6b), indicating HiCASE can detect *EGFR* mutations in plasma samples with a VAF as low as 0.13%. In summary, the HiCASE assay exhibited superior clinical performance in samples compared with ddPCR and Super-ARMS assays.

In addition, the HiCASE assay only required a minimal plasma volume. With 0.1% cfDNA, the HiCASE assay can identify L858R or 19dels using an input of 5 ng cfDNA (Supplementary Fig. 7a, b), in contrast to ddPCR which required 10 ng for L858R or 20 ng for 19dels (Supplementary Fig. 7c). In our testing of plasma samples, cfDNA extracted from 40 μL of plasma was used to detect a single mutation site. To evaluate the potential advantage of lower plasma volumes with HiCASE, we compared our method with ddPCR for detecting L858R- and 19dels-positive samples using cfDNA extracted from 40, 20, 10, 5, and 2 μL of plasma (Fig. 5a, b, Supplementary Data 3). Using the HiCASE assay, the L858R and 19dels mutations were detectable with only 10 μL of plasma in 17 samples, and even 2 μL of plasma was sufficient for the detection in four samples. In contrast, in the ddPCR assay, the L858R and 19dels mutations were detected using a 10 μL plasma volume in only 3 samples. We also evaluated the minimum cfDNA input from plasmas in samples in the 18 positive cases. For detecting the L858R mutation, it amounted to 0.14 ng in sample 6, with a VAF of 10.8%. As the 19dels mutation, this input was 0.21 ng in sample 18, corresponding to a VAF of 14.2% (Supplementary Fig. 8). We further examined whether Super-ARMS required a larger plasma volume to increase the detection efficiency. Namely, when the plasma volume increased from 40 μL to 80 μL, L858R- and 19dels-positive samples were changed from 20 to 22, indicating that the sensitivity of Super-ARMS assay less correlated with plasma volume (Supplementary Fig. 9 and Supplementary Data 4).

## Detection of *EGFR* T790M/C797S in cis/trans positions by HiCASE assay

The emergence of C797S point mutation, commonly arising with T790M. When the C797S and T790M mutations occur on the same allele in nearly all samples of C797S mutation, this is termed cis (85%), whereas their occurrence on different alleles is termed trans (15%). Notably, C797S and T790M in cis or trans position have potential therapeutic value; however, it is difficult to discern T790M/C797S mutation positions with PCR-based assay. We found that the HiCASE assay was able to clearly distinguish C797S and T790M mutation positions. The mutagenic primers were designed to generate an RE site at the 5′end while removing the original RE site at the 3′end of this sequence (Fig. 6a). Sanger sequencing showed the successful incorporation of two desired bases into the target sequence (Fig. 6b). The mixed plasmids of wild type and mutant type were pre-amplified by PCR and then digested by the Hpych4V enzyme; the wild-type fragments were completely cleaved. If the C797S and T790M mutations were in trans, only T790M mutant fragments were cleaved, leaving the C797S uncleaved. In the cis position of C797S and T790M mutations, cleavage occurred because the T790M mutation served as the RE site for the Hpych4V enzyme. As shown in Fig. 6c, the four target sequences of wild type, T790M, C797S, and T790M/C797S were cleaved by the Hpych4V enzyme, and the cleaved bands were consistent with the expected result. The size of the single C797S mutation fragments was 120 bp, while the T790M/C797S double mutation fragments were cleaved into 75 and 45 bp. We used 140 and 290 μL absolute ethanol to isolate 120 and 75 bp fragments separately for HiCASE detection (Fig. 6d). The optimal crRNA-6, which targets the C797S mutation, was selected from the pool of eight crRNAs (Supplementary Fig. 10a). By adjusting the spacer length of crRNA-6, we enabled the detection of the cleaved products of T790M-cis-C797S mutation with the Cas13a/crRNA-1 complex (Supplementary Fig. 10b). By employing the HiCASE assay, the purified products were detected, showing a clear differentiation between the C797S and T790M mutations at various positions, even at a 1% VAF (Fig. 6e), indicating the robust performance of this assay in detecting the C797S mutation.

## Discussion

ctDNA is widely considered an optimal material for obtaining tumor genetic information and guiding clinical treatment, especially in detecting genetic mutations. However, its application has large technical requirements when dealing with rare mutant copies[23,24]. Herein, we described a highly sensitive and specific method for detecting ctDNA mutations. Our method, HiCASE, exhibited high sensitivity for lower mutant allelic frequency and enabled the detection of 0.01% frequency with only 40 ng input of cfDNA standards, indicating that HiCASE can detect several mutant copies. Furthermore, we confirmed the potential of the HiCASE assay in clinical applications through the verification of 140 plasma samples. It covered approximately 90% of *EGFR* mutation sites, with sensitivity exceeding 80% and specificity of 100%. In addition, in the process of material preparation, only 200 μL plasma separated from a clinical EDTA tube is required, and the cfDNA can be extracted using an economical extraction kit instead of a costly one, lowering the overall testing expenses. The HiCASE assay involves the combination of PCR amplification, RE digestion, and detection by the Cas13a/crRNA system. The DNA cleavage capability of RE is known for its remarkable efficiency and specificity. Compared with PCR coupled with the Cas13a/crRNA system, our method significantly improved mutation detection sensitivity by 1–2 orders of magnitude, which was achieved through the RE-specific digestion of wild-type fragments. Given the absence of specific RE sites, we utilized traditional site-directed mutagenesis as an efficient means to address and resolve this issue. Indeed, this research represents the integration of CRISPR/Cas13a with RE, which has achieved notable levels of sensitivity and specificity for the detection of ctDNA mutations. Importantly, there have been few studies evaluating the clinical performance of the Cas13/crRNA ctDNA mutation assay in blood samples[25].

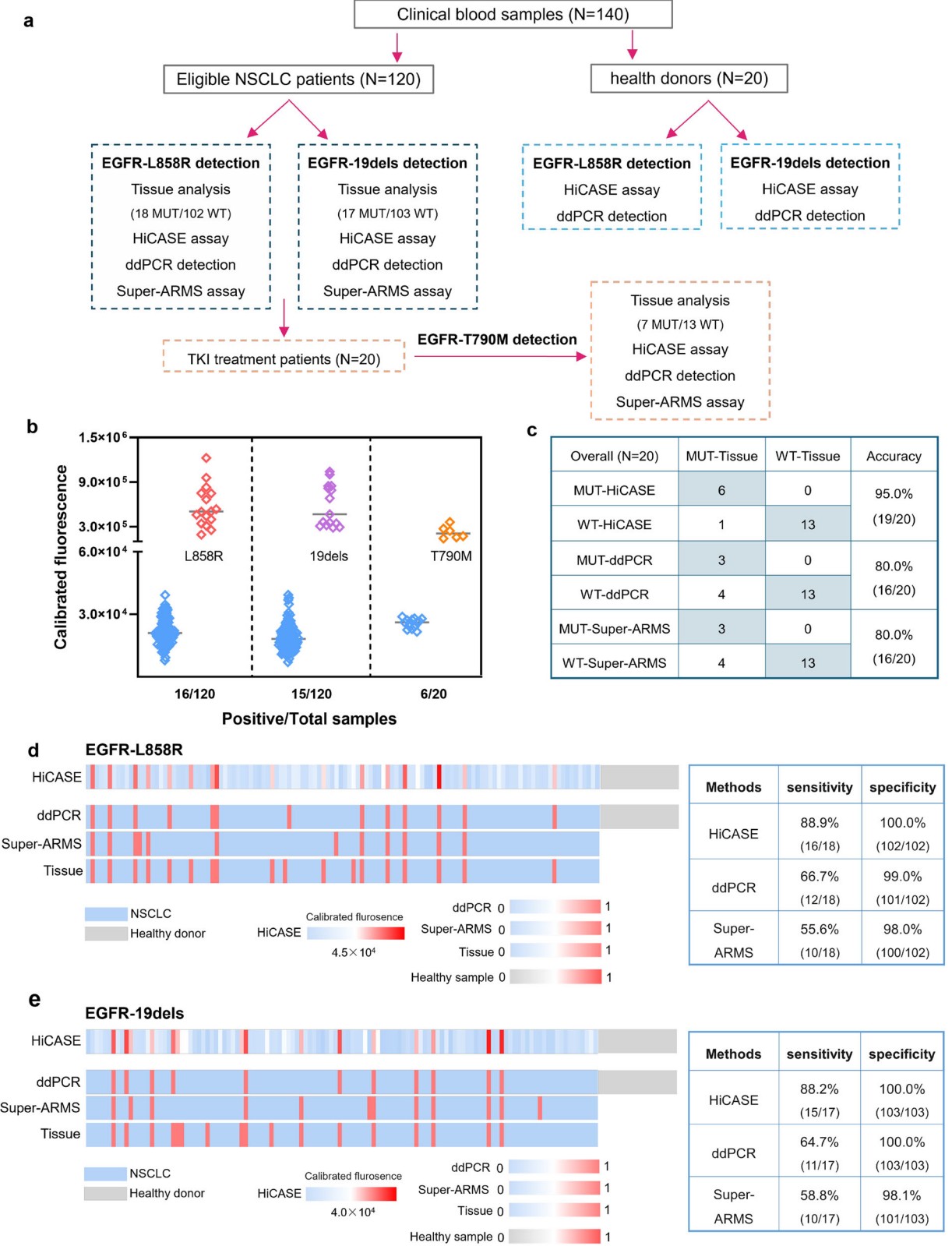

**Fig. 4 | Detecting *EGFR* mutations of blood samples using HiCASE, ddPCR, and Super-ARMS assays. a** The flowchart for detecting *EGFR* mutations of blood samples using HiCASE, ddPCR, and Super-ARMS assays. 120 blood samples from 85 adenocarcinoma and 35 squamous cell carcinomas were collected to determine the detection sensitivity and specificity of three assays. 20 blood samples from health donors were involved to verify the specificity of the HiCASE assay. **b** The detection of L858R, 19dels and T790M mutations using the HiCASE assay. **c** The confusion table of detecting result of T790M mutation in different methods. The detection of tissue samples was used as the gold standard. **d**, **e** Comparison the sensitivity and specificity of HiCASE, ddPCR, and Super-ARMS assays for detecting L858R and 19dels mutations. *n* = 3 independent experiments. The error bars indicate standard deviation.

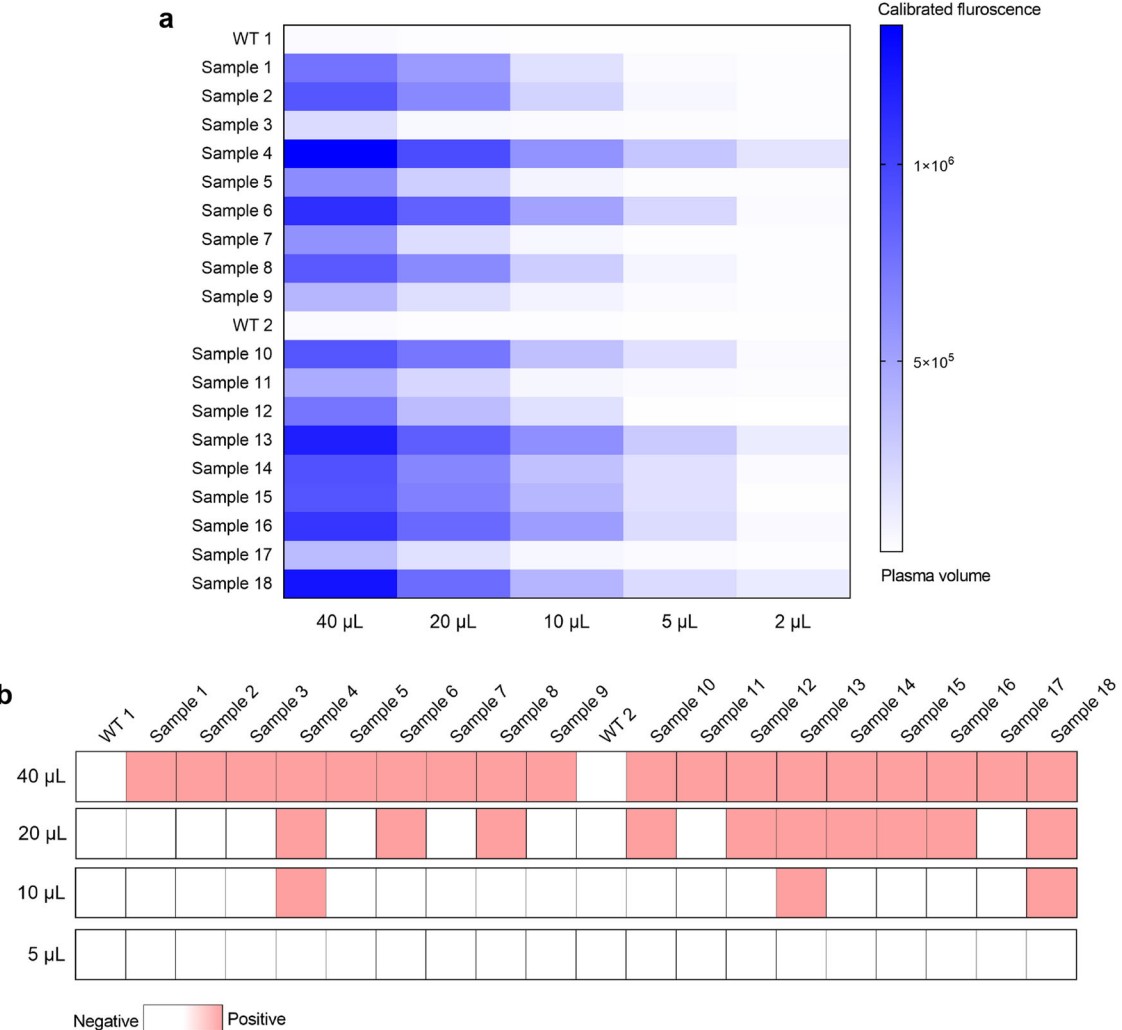

**Fig. 5 | Plasma volume needed for detecting *EGFR* mutations with HiCASE and ddPCR. a** Detecting 18 plasma samples including L858R mutation (*N* = 9) and 19dels (*N* = 9) for determining the plasma volume of HiCASE assay. **b** The 18 plasma samples deriving from (**a**) were also detected by ddPCR assay with 40 μL, 20 μL, 10 μL and 5 μL. *n* = 3 independent experiments.

Our work demonstrated the method's promise for the application in ctDNA mutations.

Numerous techniques have been developed to address the challenge of detecting ctDNA mutation sites at low frequencies. Some assay kits have obtained clinical certification, such as the Super-ARMS assay kit, which is well-suited for the detection of pathological tissues but less sensitive to ctDNA-derived mutations. In this study, we provided evidence indicating that the HiCASE assay had superior performance in detection sensitivity compared with the Super-ARMS assay. The number of positive samples in detecting *EGFR* mutations with HiCASE was greater than that with the Super-ARMS assay (37 vs. 23). Our results also showed that the HiCASE assay exhibited superior sensitivity compared with ddPCR (88.1 vs. 61.9%), indicating the advantage of HiCASE for evaluating *EGFR* mutants in ctDNA. Besides, we illustrated that HiCASE needed a lower volume of plasma for detecting ctDNA mutations than did ddPCR, which was related to the high sensitivity of our method. Some highly sensitive detection technologies for ctDNA have been reported. Liu et al. introduced a method using the thermophilic PfAgo protein coupled with DNA polymerase to detect ctDNA mutations in a single tube, achieving a sensitivity of 0.1%[26]. Certainly, pre-amplification of

cfDNA is also a requirement before employing this method. Xu et al. developed the specific terminal mediated PCR method for detecting L858R but exhibited a sensitivity of 1%[10]. It further demonstrated that the HiCASE assay has ultrasensitive detection for ctDNA mutation.

In this study, we discovered that HiCASE shows promise in detecting ctDNA mutations in early-stage lung cancer. Among cases with *EGFR* mutations, approximately 48% (15/31) of NSCLC patients exhibited thoracic confinement, suggesting that these individuals could potentially benefit from early surgical intervention or targeted drug therapy to enhance survival rates. It is noteworthy that our method can clearly distinguish between different mutated positions of T790M and C797S, providing valuable guidance for targeted drug applications[27,28]. It is difficult to determine the mutated positions of T790M and C797S using conventional PCR based methods. Some literature has also reported that ddPCR could be employed to detect T790M and C797S mutations by observing fluctuations in the amplitude of fluorescent signals[29,30]. Nevertheless, the disparity in amplitude between T790M and C797S is less evident, particularly at lower mutation frequencies. In our approach, the HiCASE assay exhibited enhanced performance in discerning T790M/C797S in cis/trans positions even when the VAF of C797S was as low as 1% or less.

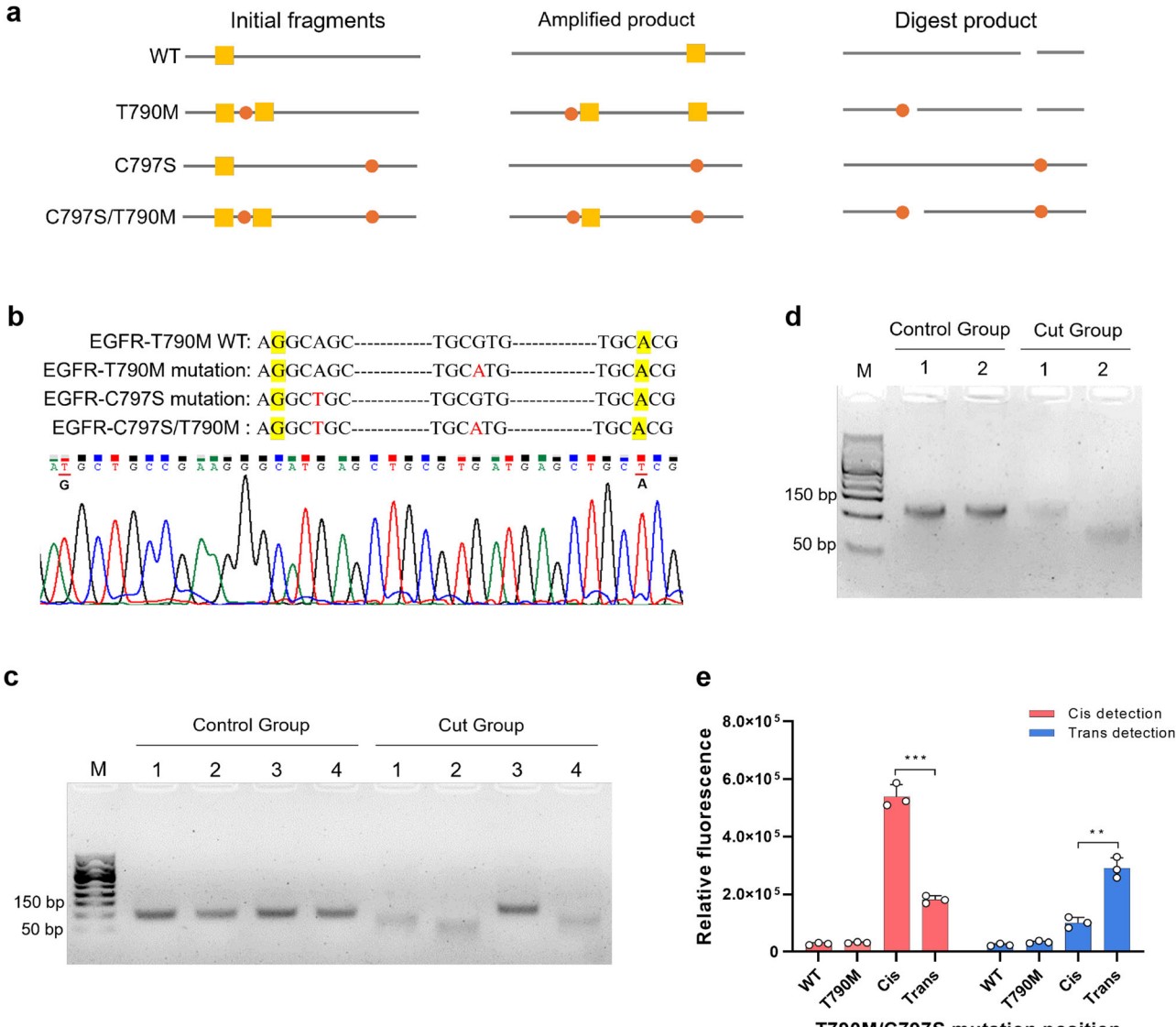

**Fig. 6 | Detecting T790M and C797S mutations in different positions using the HiCASE assay. a** Schematic diagram of restriction enzyme sites formed after PCR amplification of wild-type, T790M, C797S, and T790M-cis-C797S mutation. Red indicated the mutated base of wild type and yellow for RE site. **b** The result of Sanger sequencing for *EGFR* C797S mutation after site-directed mutagenesis. Yellow indicated the base to be mutated and red for mutated base of wild type. **c** The electropherogram of PCR products of T790M/C797S before and after digesting by the Hpych4V enzyme. (1) T790M/C797S wild type; (2) T790M mutation; (3) C797S mutation; (4) T790M-cis-C797S mutation; **d** The purified products of T790M/C797S with cis/trans fragments after digestion. (1) T790M-cis-C797S fragments; (2) T790M-trans-C797S fragments; **e** The detection of T790M and C797S cis/trans mutation with HiCASE assay. *n* = 3 independent experiments, two-tailed Student's *t* test, The error bars indicate standard deviation; *$P < 0.05$, **$P < 0.01$, ***$P < 0.001$, and ****$P < 0.0001$.

The HiCASE assay also presents some limitations that will require improvement in the future. For example, to reduce the background signal caused by wild type, we incorporated the RE to selectively cleave wild-type products, thereby increasing the operational steps in the detection process. Indeed, the HiCASE assay (140–160 min) takes a similar time with ddPCR (130–140 min) and ARMS-PCR (120 min). In addition, we used 120 plasma samples from NSCLC patients to evaluate the clinical utility of HiCASE, and the presence of *EGFR* mutations was confirmed in 37 samples. However, the 20 plasma samples were insufficient to assess the T790M mutation, even though HiCASE assay determined a greater number of positive samples compared with ddPCR and Super-ARMS assays.

In conclusion, we developed an exceptionally sensitive and specific assay for the detection of *EGFR* mutations. We successfully validated the outstanding clinical utility of HiCASE using a group of clinical blood samples. Certainly, the HiCASE assay is also well-suited for the detection of driver genes other than *EGFR*, suggesting its considerable promise for detecting ctDNA mutations.

## Methods
### Study design and clinical samples
The sample size of 140 blood samples were collected from 120 NSCLC patients (85 lung adenocarcinomas, 35 lung squamous cell carcinoma) and 20 healthy donors between May 2019 and December 2020. The inclusion criteria were as follows: (1) Eligible subjects were men or women who were >18 years of age; (2) Patients with pathologically diagnosed NSCLC; (3) Participants with an adequate amount of peripheral blood samples. Exclusion criteria: (1) Insufficient blood volume; (2) Presence of severe autoimmune diseases; (3) Lack of tissue *EGFR* mutation detection results. Healthy volunteers were recruited to participate as negative controls. All participants with written informed consent. This study was approved by the

Ethics Review Committee at First People's Hospital of Chenzhou (No. 20190517). All ethical regulations relevant to human research participants were followed.

cfDNA was extracted from plasma samples following the steps below: about 1 mL plasma was separated from blood by centrifugation at 2500 $g$ for 10 min at 4 °C and continued to centrifugate at 12,000 $g$ for 10 min at 4 °C. The cfDNA extracted kit was using a VAHTS® Serum/Plasma Circulating DNA Kit (Vazyme, Nanjing, China, Cat# N902) according to the manufacturer's instructions. Finally, the cfDNA was eluted with 100 μL RNase-free water and stored at −20 °C.

## Nucleic acid preparation
The wild-type and mutant-type plasmids were synthesized by genscript company (Nanjing, China). All the mutant sites were obtained from COSMIC (https://cancer.sanger.ac.uk/signatures/signatures_v2/) database version 2. Different VAFs were mixed as described previously[31]. Briefly, mutant-type and wild-type templates were combined at varying ratios of 10, 1, 0.1, and 0.01%. The concentration of the wild-type template was maintained at 0.01 ng/μL, while the mutant-type templates were diluted in a gradient ranging from 0.01 to 0.000001 ng/μL. The DNA templates were stored at −20 °C.

The transcription templates of crRNA were amplified using a universal primer containing the T7 promoter and a specific primer containing the spacer sequence. Purified PCR products containing the T7 promoter were transcribed into crRNA using the transcriptAid T7 High Yield Transcription Kit (Thermo Fisher Scientific, Waltham, MA, USA, Cat# K0441) according to the manual. The transcribed crRNA was purified by the miRNeasy Micro Kit (QIAGEN, Hilden, Germany, Cat# 217084) and stored at −80 °C. The MUT-specific crNRA with high efficiency was screened out by in vitro cleavage of Cas13a and fluorescent cleavage assay. All crRNA sequences are shown in Supplementary Data 5.

## In vitro Cas13a cleavage
The Cas13a cleavage reaction consisted of 25 mM Tris HCl (PH 7.4), 9 mM MgCl$_2$, 200 ng Cas13a (Genscript, Cat# Z03486), 200 ng crRNA, 500 ng target RNA, and DEPC water added to the total mixture volume of 20 μL. The mixture was incubated at 37 °C for 30 min and heated at 70 °C for 10 min to denature RNA. The cleavage mixture was loaded onto 12% TBE-Urea Gels with 120 V for 60 min and analyzed on a gel imager system (Thermo Fisher Scientific). As the fluorescent cleavage assay, the Cas13a/crRNA reaction included 25 mM Tris HCl, 9 mM MgCl$_2$, 100 ng Cas13a, 20 ng crRNA, 10 mM rNTP (NEB, Shanghai, China, Cat# N0466S), 15 U T7 RNA polymerase (NEB, Cat# M0251S), 8 U RNA inhibitor (NEB, Cat# M0314S), 0.25 μM FAM-labeled ssRNA reporters, 20 ng DNA template, 50 ng total human RNA (Thermo Fisher Scientific, Cat# 4307281), and DEPC water added to the total mixture volume of 20 μL. The fluorescent signal of the Cas13a reaction was detected by the 7500 fast Real-Time PCR Systems (Thermo Fisher Scientific, USA) at 37 °C for 40 min and analyzed with the 7500 Fast Software v2.3. By observing the TBE-Urea gels and fluorescence values, we determined the activity and specificity of the mutant-crRNA.

## Detection of *EGFR* mutations by HiCASE
In brief, the cell free DNA (cfDNA) was initially subjected to PCR amplification, followed by treatment with an RE that specifically cleaved wild-type fragments, thereby efficiently eliminating background interference from the wild-type fragments. Subsequently, the digested product was detected with the Cas13a/crRNA system. To evaluate the sensitivity of HiCASE detection system, mutant-type, and wild-type plasmids were mixed at different VAFs, including 10, 1, 0.1, and 0.01%. The wild-type plasmid was maintained at $10^{-3}$ ng/μL, while the mutation plasmids were gradually added in the range from $10^{-4}$ to $10^{-8}$ ng/μL. Different VAFs of plasmids were amplified by PCR with a 30 μL reaction system containing 1×Taq mix, 0.5 μM of primers, 1 μL of plasmids, and DEPC H2O. The PCR amplification condition consisted of

pre-incubation at 95 °C for 5 min; 36 cycles of 95 °C for 30 s, 56 °C for 15 s, and 72 °C for 30 s; and holding at 72 °C for 5 min. About 150 ng of PCR products were digested by RE at 37 °C for 40 min. Then, the digested products were purified using the Select-a-Size DNA Clean & Concentrator Kit. Finally, they were detected by Cas13a/crRNA reaction for 40 min, The reaction consisted of 25 mM Tris HCl, 9 mM MgCl$_2$, 100 ng Cas13a, 5 nM crRNA transcript template, 10 mM rNTP, 15 U T7 RNA polymerase, 8 U RNA inhibitor, 0.25 μM FAM-labeled ssRNA reporters, digested products, 50 ng total human RNA, and DEPC H2O added to the total mixture volume of 20 μL. The fluorescent signals were collected with Real-Time PCR Systems.

To establish a standardized HiCASE detection system for clinical sample analysis, cfDNA Reference Standards (Horizon Discovery, Cambridge, UK, Cat# HD825) were purchased from a commercial supplier. cfDNA standards consisted of *EGFR* del15, L858R, and T790M with different VAFs (0.1, 1, and 5%) at a concentration of 20 ng/μL. To achieve VAFs of 0.2 and 0.02%, we mixed 16 ng of wild-type DNA with 4 ng of mutant-type cfDNA at 1% and 16 ng of wild-type DNA with 4 ng of mutant-type cfDNA at 0.1%. Different VAFs (0.1, 0.2, 1, 0.2, and 5%) cfDNA standards were amplified with a 30 μL reaction system including 15 μL of 2×Taq mix, 0.5 μM of primers, 1 μL of cfDNA standard, and DEPC H2O. The PCR cycling program started with a pre-incubation at 95 °C for 5 min; 36 cycles of 95 °C for 30 s, 56 °C for 15 s, and 72 °C for 30 s; and holding at 72 °C for 5 min. About 150 ng of PCR products were digested by RE at 37 °C for 40 min. The reaction system consists of 5 μL of 10× Cutsmart buffer, 10 U RE, 150 ng DNA, and adding DEPC H2O to a final volume of 50 μL. Following this, the digested products were purified, added to the Cas13a/crRNA reaction, and incubated for 40 min, The reaction consisted of 25 mM Tris HCl, 9 mM MgCl$_2$, 100 ng Cas13a, 5 nM crRNA transcript template, 10 mM rNTP, 15 U T7 RNA polymerase, 8 U RNA inhibitor, 0.25 μM FAM-labeled ssRNA reporters, digested products, 50 ng total human RNA, and DEPC H2O added to the total mixture volume of 20 μL. The FAM fluorescent signal was acquired using the Real-Time PCR Systems.

For identifying *EGFR* mutations in plasma samples, 2 μL of extracted cfDNA from plasma was detected by HiCASE. The operational process was consistent with the standardized HiCASE detection system. The detection process for all clinical samples follows this protocol. Besides, the cfDNA extracted from 40, 20, 10, 5, and 2 μL of plasma was employed for detection using the HiCASE assay.

## Detection of *EGFR* mutations by ddPCR
ddPCR was used to detect *EGFR* dels, L858R and T790M mutations. The reaction mixture consisted of 1 × ddPCR supermix (Bio-Rad, Shanghai, China, Cat# 186-3024), 0.4 μM wild-type and mutant-type probes, 0.5 μM primers, 20 ng cfDNA standard, and DEPC H$_2$O. In the examination of plasma samples, 2 μL extracted cfDNA was used to detect a single mutation site. For testing different volumes of plasma, cfDNA extracted from 40 μL, 20 μL, 10 μL, and 5 μL of plasma were subjected to ddPCR, respectively. The mixture was added to the QX200 droplet generator (Bio-Rad, Cat# 17005227) with 70 μL of oil (Bio-Rad, Cat# 1863005), and then it was transferred to a 96-well PCR plate (Bio-Rad, Cat# 12001925) and sealed with a sheet of tinfoil (Bio-Rad, Cat# 1814040). The PCR was carried out as follows: pre-incubation at 95 °C for 5 min, 40 cycles of 95 °C 30 s and 60 °C 1 min, 98 °C for 10 min. The PCR products were analyzed using QX200 Droplet Reader (Bio-Rad, Cat# 17005228) and QuantaSoft Software version 1.7.4 according to the manual. The positive threshold value was determined to be more than 2 mutation droplets.

## Fluorescence data analysis of HiCASE
To calculate background-subtracted fluorescence data, the relative fluorescence of each sample was obtained from the fluorescence difference

between the sample and negative control as follows:

$$\text{Relative fluorescence} = (Send - Son) - (Nend - Non),$$

where $S_{end}$ and $S_{on}$ represent the final fluorescence and initial fluorescence of the sample, respectively, and $N_{end}$ and $N_{on}$ represent the final fluorescence and initial fluorescence of the negative control, respectively. The detected cfDNA fluorescence of the plasma samples was calibrated by cfDNA standards as follows:

$$\text{Calibrated fluorescence}: \frac{x1}{x2} = \frac{y1}{y2},$$

where χ1 refers to the relative fluorescence of 1% cfDNA standard in this new experiment, χ2 refers to the relative fluorescence of 1% cfDNA standard in established cfDNA standards systems, y1 indicates the relative fluorescence of the sample in this new experiment, and y2 indicates the calibrated fluorescence of the sample.

### Statistics and reproducibility
Data were expressed as mean ± SD. and all statistical tests are two-sided, analyzed with Student's t-test using Graph Pad Prism 9.0 (Inc., La Jolla, CA, USA). $P < 0.05$ was regarded as statistically significant. Every experiment was repeated three times independently.

### Reporting summary
Further information on research design is available in the Nature Portfolio Reporting Summary linked to this article.

### Data availability
Supplementary Figs. and Supplementary Methods are included in Supplementary materials. All source data used for generating graphs and charts in main figures are deposited in Supplementary Data 6. Any other data is available from the corresponding authors on reasonable request.

### Code availability
Details of publicly available software used in the study are given in the "Methods". No custom code or mathematical algorithm that is deemed central to the conclusions was used.

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

## Acknowledgements

This work was supported by National Natural Science Foundation of China [82372344], Natural Science Foundation of Guangdong Province [2023A1515011925]. Shenzhen Science and Technology Innovation Program [JSGG20210802153410031], Shenzhen Nanshan District Science and Technology Plan Project [NS016], Huazhong University of Science and Technology Shenzhen Hospital Research Leaders Start Fund [YN2021002], Daan Gene Horizontal Project [HXKY2022002], Shenzhen University Medical-Engineering Interdisciplinary Research Fund Project [2023YG022], International Science and Technology Independent Cooperation Project [GJHZ20220913144213025], Newly introduced discipline leader fund project in Nanshan District of Shenzhen City [NSZD2023020], High-level hospital health science and technology project [NSZD2023040].

## Author contributions

D.L., T.L. and Q.L. conceived the original idea. D.L., T.L., L.W., and X.W. designed the experiments. L.W. and X.W. performed most of the experiments, data analysis, and data visualization. Y.Y. and J.W. prepared reagent supplies and DNA templates. Z.H., J.J., L.D., X.L. collected blood samples, clinical data and extracted cfDNA. Y.H. and Y.L. prepared crRNAs of Cas13a. Z.L. and X.Z. checked and screened potentially eligible experiments. L.W. and X.W. wrote the draft manuscript. D.L., T.L., and Q.L. revised the manuscript.

## Competing interests

The authors declare no competing interests.
