## [Peer Review File · Communications Biology]

Reviewers' comments:

Reviewer #1 (Remarks to the Author):

In this study, Authors provide a new technology for the detection of EGFR mutations in the circulating tumor DNA, showing remarkable sensitivity and specificity.

In the abstract, Authors should be more factual and provide less comments on their novel technique. Examples "sensitivity of 0.01% only...", "obviously outperforming".

"Simultaneously, our research effectively facilitated translation from basic to clinical" This statement is not really adapted at the conclusion of such an abstract.

"Nevertheless, peripheral blood harbors low-frequency mutations of ctDNA surrounded by the tremendous wild-type DNA shed from normal cells, challenging the detection of ctDNA mutations in blood with conventional technologies". Please remove "tremendous". There are several limitations to the success of ctDNA sequencing, and authors should mention that the variable DNA shedding from tumors is among one of the most important, especially in NSCLC.

"recognized for its exceptional efficiency": moderate the "exceptional".

I suggest Authors to briefly mention the main aspects of their novel technology in the introduction, moving its long and complete description in the Methods.

"To the best of our knowledge, this research represents the first instance in which the integration of CRISPR/Cas13a with RE has achieved such remarkable levels of sensitivity and specificity for the detection of ctDNA mutations. Importantly, there have been few studies evaluating the clinical performance of the Cas13/crRNA ctDNA mutation assay in blood samples. Our work demonstrated the method's substantial promise for the application in ctDNA mutations." I suggest Authors to remove this paragraph, moving it eventually to the discussion.

In the section "Detection for cfDNA extracted from NSCLC plasma samples with different assays", Authors should specify how many samples from patients with NSCLC harbored EGFR mutations in the tissue analysis, and providing in the text the respective incidence of specific EGFR mutations. The same numbers should be provided in figure 4a, to allow a complete understanding of the results of ctDNA sequencing.

If I understand correctly Figure 4d and 4e, n = 35 patients were EGFR+ NSCLC. In addition to the % of specificity and sensitivity, the fractions should be provided in the text (18 as a denominator for L858R, 17 as a denominator for del19).

One major element of interest in NSCLC is the low DNA shedding in plasma, especially in limited disease stages and when diseases are limited to the thorax and brain. Can Authors provide details of the tumor spread at the moment of plasma sampling, and correlate that with the performance of their new assay? This would be of great interest from a clinical angle.

In the discussion, Authors can strengthen the clinical relevance of their assay as technologies with high sensitivity/specificity for EGFR mutations detection are of crucial importance in patients with early-stage NSCLC (ie disease limited to the thorax) that undergo surgical treatment. Osimertinib is the standard of care in the adjuvant setting of these patients, and additional predictive biomarker are under investigation. A reliable technology as the one here presented could be applied for this scope.

Reviewer #2 (Remarks to the Author):

This well written manuscript by Wang and colleagues introduces a novel method for detecting cell-free DNA in plasma, utilizing CRISPR/Cas and restriction enzymes. Through the integration of these two techniques, their newly proposed method demonstrates the capability to detect mutated cell free DNA in patient plasma samples with remarkable sensitivity—a crucial attribute for this application. The methodology is described in detail, and from my evaluation, it appears that all necessary controls have been implemented. The presented method undergoes validation using established NSCLC samples and surpasses the performance of conventional ddPCR, requiring even less input. Furthermore, its potential application extends beyond NSCLC, making it a promising tool for detecting various mutations. While the manuscript is commendable, I do have one major concern and a few minor ones, which I am confident the authors can address:

Major:

1) The HiCASE method presented in this study undergoes a comprehensive comparison with ddPCR and Super-ARMS, both theoretically and in practical detail, as outlined in the results section. The authors only briefly touch upon NGS-based methods in the introduction (line 61), characterizing them as time-consuming and limited. However, it is essential to acknowledge the evolving prominence of NGS-based approaches. These methods, applicable in WGS/WES scenarios without bias for a specific target region, or through targeted approaches to achieve very high depth without incurring excessive costs (e.g., CAPP-Seq), offer advantages. NGS-based methods can be standardized, semi-automated, and rapidly scaled up for multiple mutation locations, thereby exhibiting flexibility and competitiveness. This is particularly noteworthy as sequencing costs continue to decline. It would be valuable if the authors could consider including a comparison of their method with, for instance, CAPP-Seq in their analysis.

Minor:

1) I appreciate the authors' attention to the input DNA requirement, particularly as HiCASE necessitates less input. It would be insightful to determine the minimum required DNA amount (in addition to plasma volume).

2) Developing a specific crRNA for each new mutation is a crucial aspect of the assay. The authors address this challenge by designing a range of crRNA (up to 8) for each site and testing their specificity and sensitivity. Do the authors consider this as a significant drawback of the assay, given that the design and testing process is not as straightforward as, for example, primers for PCR or the design for CAPP-Seq sites?

3) Regarding Fig. 2 h-i: I suggest modifying the legends of the heatmap to avoid displaying a spectrum between 0 and 1, as only 0 and 1 can be called. Adding explanations for what 0 and 1 represent in the legend would enhance readability.

4) In Fig. 4 c-e, the legends mention $n=3$, but the actual figures (e.g., Fig. 4C) indicate different numbers. Shouldn't the n value for Fig. 4d-e be much higher, considering the inclusion of 120 NSCLC patients and 20 healthy donors?

Reviewer #3 (Remarks to the Author):

In the manuscript "CRISPR/Cas13a based supersensitive circulating tumor DNA assay for 1 detecting EGFR mutations in plasma" Wang and colleagues combine CRISPR/Cas13a based mutation detection

with selective digestion of WT fragments through restriction enzymes to allow sensitive EGFR mutation detection in NSCLC. Custom assays/workflows were generated for L858R, T790M, C797S and exon 19 deletions. The authors benchmark their assays with synthetic DNA partially achieving sensitivities up to 0.01% before applying their method to 120 patient plasma samples outperforming ddPCR and Super-ARMS.

The manuscript is of interest and the research question timely. Yet, the manuscript has significant weaknesses related to the presentation of the data and provided methodologic details which precludes publication in its current form. Specific comments to improve the manuscript are below.

- Value of restriction enzyme digestion: The key advance of the HiCASE method presented here over other CAS based methods like SHERLOCK is the combination of CRISPR/Cas13a based mutation detection with selective digestion of WT fragments through restriction enzymes (RE). However, the authors fail to convincingly show that the incorporation of RE improves detection. While Fig 2C does show an advantage for L858R detection when using RE digestion, I find it remarkable that the overall fluorescence signals for mutant containing conditions go up and only the WT-only signal goes down. The authors state that the RE step is supposed to reduce the background, but it seems to selectively do so in the condition with only WT fragments. Do the authors have an explanation for that? The authors should show similar experiments for other mutations, and also demonstrate the advantage of the RE step in primary patient samples.

- Methodology:

1. There seems to be a huge discrepancy between the synthetic DNA testing experiments, and the experiments around patient plasma samples. For the patient plasma samples only 40 μ l input plasma were used. Why do the authors not refer to masses as in the remainder of the manuscript? With plasma cfDNA concentrations usually being around 10ng/mL, 40 μ l of plasma would only account for 0.4ng DNA reflective of 100-200 tumor genomes as input. In contrast for synthetic DNA 20-40 ng was used, which seems vastly different. The authors should report the input DNA mass used for each of the 120 patient samples.

2. What was the mean AF in each of the 120 patient samples. In order to detect mutations from just 100-200 genomes (see above), the ctDNA concentration must be really high. Orthogonal data by conventional sequencing must be provided in order to make the data interpretable.

3. I also did not really understand what kind of synthetic DNA was used for the technical experiments. The authors partially refer to plasmids (circular?) or cfDNA standards. The authors should clarify on fragment length and whether the synthetic DNA (WT and mutant) only covered the EGFR locus or the full genome. This is particularly important since the dilutions were apparently made by mass, and the mass of a full genome is >> larger than the mass of a copy of EGFR.

4. Description of methods is very sparse especially given the authors aim to introduce a new method here. The authors need to add details on their precise workflow when starting from synthetic DNA as well as patient samples. Was the same number of PCR cycles done for technical experiments and for the patient samples? Was synthetic DNA amplified at all? Given the high number of PCR cycles reported in the methods section (36 cycles!) I would expect to see a significant background through the introduction of error through PCR errors which would likely preclude sensitive detection at levels of 0.01%.

- Manuscript presentation: The manuscript is partially hard to follow. For instance, the visualization of introduced RE sites needs to be improved. In the current depiction, it is unclear how the mutant primers +/- EGFR mutations lead to a specific RE site.

Minor comments:

- Fig 4c is not interpretable since ground truth information is not provided. Should be replaced through confusion matrices.

- Supp. Fig 4. Every input mass needs a WT control since it is expected to impact background.

Response to the Editor and Reviewers:

Thank you for your letter and for the reviewers' comments concerning our manuscript entitled "CRISPR/Cas13a-based supersensitive circulating tumor DNA assay for detecting EGFR mutations in plasma". Those comments are valuable and very helpful for revising and improving our paper.

Reviewer 1 (Minor or Recommended Revisions)

Comment 1: In the abstract, Authors should be more factual and provide less comments on their novel technique. Examples "sensitivity of 0.01% only...", "obviously outperforming". "Simultaneously, our research effectively facilitated translation from basic to clinical" This statement is not really adapted at the conclusion of such an abstract.

Response: Thank you for your advice. We removed the "only" and "Simultaneously, our research effectively facilitated translation from basic to clinical", and changed "obviously outperforming" to "higher than" in the Abstract section. The Abstract section revised as follows:

~~Detecting gene mutations in circulating tumor DNA (ctDNA) is crucial in clinical applications.~~ Despite recent technological advancements in ctDNA mutation detection, challenges persist in identifying low-frequency mutations due to inadequate sensitivity and coverage of current procedures. Herein, we introduced a super-sensitivity and specificity technique for detecting ctDNA mutations, named HiCASE. The method utilizes PCR-based CRISPR, coupled with the restriction enzyme. In this work, HiCASE focused on testing a series of *EGFR* mutations to provide enhanced detection technology for non-small cell lung cancer (NSCLC), enabling a detection sensitivity of 0.01% ~~only~~ with 40 ng cell free DNA (cfDNA) standard. When applied to a panel of 140 plasma samples from 120 NSCLC patients, HiCASE exhibited 88.1% clinical sensitivity and 100% specificity with 40 μ L of plasma, ~~obviously outperforming~~ **higher than** ddPCR and Super-ARMS assay. In addition, HiCASE can also clearly distinguish T790M/C797S mutations in different positions at a 1% variant allele frequency (VAF), offering valuable guidance for drug utilization. Indeed, the established HiCASE assay showed significant potential for clinical applications. ~~Simultaneously, our research effectively~~

~~facilitated translation from basic to clinical.~~

Comment 2: “Nevertheless, peripheral blood harbors low–frequency mutations of ctDNA surrounded by the tremendous wild-type DNA shed from normal cells, challenging the detection of ctDNA mutations in blood with conventional technologies”. Please remove “tremendous”. There are several limitations to the success of ctDNA sequencing, and authors should mention that the variable DNA shedding from tumors is among one of the most important, especially in NSCLC.

Response: Thank you for your advice. In the revised Introduction section, we changed “Nevertheless, peripheral blood harbors low–frequency mutations of ctDNA surrounded by the tremendous wild-type DNA shed from normal cells, challenging the detection of ctDNA mutations in blood with conventional technologies” to “Nevertheless, in the early and mid stages of lung cancer, the variable DNA shedding from tumor is relatively low and surrounded by the wild-type DNA shed from normal cells. Moreover, ctDNA displays highly fragmented characteristics and a short half-life, further complicating detection with conventional technologies” (in line 51).

Comment 3: “recognized for its exceptional efficiency”: moderate the “exceptional”.

Response: Thank you for your advice. We changed “recognized for its exceptional efficiency” to “recognized for its high efficiency”.

Comment 4: I suggest Authors to briefly mention the main aspects of their novel technology in the introduction, moving its long and complete description in the Methods.

Response: Thank you for your advice. We moved the Introduction content of “In brief, the cell free DNA (cfDNA) was initially subjected to PCR amplification, followed by treatment with an RE that specifically cleaved wild-type fragments, thereby efficiently eliminating background interference from the wild-type fragments. Subsequently, the digested product was detected with the Cas13a/crRNA system” to Methods (in line 372), marked in yellow.

Comment 5: “To the best of our knowledge, this research represents the first instance in which the integration of CRISPR/Cas13a with RE has achieved such remarkable levels of sensitivity

and specificity for the detection of ctDNA mutations. Importantly, there have been few studies evaluating the clinical performance of the Cas13/crRNA ctDNA mutation assay in blood samples. Our work demonstrated the method's substantial promise for the application in ctDNA mutations." I suggest Authors to remove this paragraph, moving it eventually to the discussion.

Response: Thank you for your advice. We moved this paragraph to the discussion (in line 271).

Comment 6: In the section "Detection for cfDNA extracted from NSCLC plasma samples with different assays", Authors should specify how many samples from patients with NSCLC harbored EGFR mutations in the tissue analysis, and providing in the text the respective incidence of specific EGFR mutations. The same numbers should be provided in figure 4a, to allow a complete understanding of the results of ctDNA sequencing. If I understand correctly Figure 4d and 4e, n = 35 patients were EGFR+ NSCLC. In addition to the % of specificity and sensitivity, the fractions should be provided in the text (18 as a denominator for L858R, 17 as a denominator for del19).

Response: Thank you for your suggestion. In the revised manuscript, we have added the number and incidence rates of EGFR mutations, including L858R, 19dels, and T790M, in the tissue analysis results section and updated **Fig 4a** accordingly. Additionally, we have incorporated fractions in this result section for clarity and enhancement.

The revised result section was "To test the clinical sensitivity and specificity of the HiCASE assay, 140 plasma samples of NSCLC patients and healthy donors were collected as shown in Fig 4a. In this study, the plasma samples from the NSCLC patients were paired with tissue samples, and the analysis of *EGFR* mutation in tumor tissue was considered the gold standard.

Among the 120 NSCLC patients, 35 cases had EGFR mutations, including 18 with the L858R mutation and 17 with the 19dels mutation. In the subset of 20 patients treated with TKIs, 7 cases exhibited the T790M mutation. The incidence rates for L858R, 19dels, and T790M were 15% (18/120), 14.2% (17/120), and 35% (7/20), respectively. The HiCASE assay was employed to identify *EGFR* mutations of L858R and 19dels (Fig. 4b). We found that the negative and positive samples were clearly distinguishable based on their fluorescent values. Analysis of ctDNA revealed L858R in 13.3% (16/120), 19dels in 12.5% (15/120), and T790M in 30% (6/20)

of plasma samples. We further compared the clinical performance of HiCAGE with ddPCR and Super-ARMS assays for *EGFR* T790M mutations (Fig. 4c). The number of positive samples determined by HiCAGE, ddPCR, and Super-ARMS assays was 6, 3, and 3, respectively. As for the L858R mutation, HiCAGE, ddPCR, and Super-ARMS assays demonstrated sensitivity of 88.9% (16/18), 66.7% (12/18), and 55.6% (10/18), with specificity of 100% (102/102), 99.0% (101/102), and 98.0% (100/102), respectively. For the 19dels mutation, the sensitivity and specificity of the three assays were 88.2% (15/17) and 100% (103/103), 64.7% (11/17) and 100% (103/103), and 58.8% (10/17) and 98.1% (101/103), respectively (Fig 4d–e and Supplementary Table 1). The accuracy of *EGFR* mutations detected by HiCAGE, ddPCR, and Super-ARMS assays was 98.1% (255/260), 93.5% (243/260), and 91.2% (237/260), respectively (Supplementary Table 2). In summary, the HiCAGE assay exhibited superior clinical performance in samples compared with ddPCR and Super-ARMS assays.” The added content was marked in yellow.

Figure 4. Detecting *EGFR* mutations of blood samples using HiCAGE, ddPCR, and Super-ARMS assays. **a** The flowchart for detecting *EGFR* mutations of blood samples using HiCAGE, ddPCR and Super-ARMS assays. 120 blood samples from 85 adenocarcinoma and 35 squamous cell carcinomas were collected to determine the detection sensitivity and specificity of three assays. 20 blood samples from health donors were involved to verify the specificity of the HiCAGE assay. **b** The detection of

L858R, 19dels and T790M mutations using the HiCASE assay. **c** The confusion table of detecting result of T790M mutation in different methods. The detection of tissue samples was used as the gold standard.

d-e Comparison the sensitivity and specificity of HiCASE, ddPCR and Super-ARMS assays for detecting L858R and 19dels mutations. $n = 3$ technical replicates. Error bars represent the mean \pm S.D.

Comment 7: One major element of interest in NSCLC is the low DNA shedding in plasma, especially in limited disease stages and when diseases are limited to the thorax and brain. Can Authors provide details of the tumor spread at the moment of plasma sampling, and correlate that with the performance of their new assay? This would be of great interest from a clinical angle.

Response: Thank you for your suggestion. We collected clinical information from 120 NSCLC patients, among whom 31 cases presented with pulmonary metastasis and 30 exhibited intrapulmonary metastases, primarily involving the bone, brain, and liver. About 48% (15/31) of NSCLC patients with *EGFR* mutations were confined to the thorax and 35% (11/31) of *EGFR* mutation patients experienced bone or brain metastasis. It likely indicates that *EGFR* mutations can be detected in lung cancer before distant metastasis occurs. However, due to sample limitations, definitive conclusions cannot be obtained from this study.

Comment 8: In the discussion, Authors can strengthen the clinical relevance of their assay as technologies with high sensitivity/specificity for EGFR mutations detection are of crucial importance in patients with early-stage NSCLC (ie disease limited to the thorax) that undergo surgical treatment. Osimertinib is the standard of care in the adjuvant setting of these patients, and additional predictive biomarker are under investigation. A reliable technology as the one here presented could be applied for this scope.

Response: Thank you for your advice. In the revised manuscript, we added the content: “In this study, we discovered that HiCASE shows promise in detecting ctDNA mutations in early-stage lung cancer. Among cases with *EGFR* mutations, approximately 48% (15/31) of NSCLC patients exhibited thoracic confinement, suggesting that these individuals could potentially benefit from early surgical intervention or targeted drug therapy to enhance survival rates,”(in line 294) marked in yellow at the discussion section.

Reviewer 2 (Major or Minor Revisions)

Comment 1: The HiCASE method presented in this study undergoes a comprehensive comparison with ddPCR and Super-ARMS, both theoretically and in practical detail, as outlined in the results section. The authors only briefly touch upon NGS-based methods in the introduction (line 61), characterizing them as time-consuming and limited. However, it is essential to acknowledge the evolving prominence of NGS-based approaches. These methods, applicable in WGS/WES scenarios without bias for a specific target region, or through targeted approaches to achieve very high depth without incurring excessive costs (e.g., CAPP-Seq), offer advantages. NGS-based methods can be standardized, semi-automated, and rapidly scaled up for multiple mutation locations, thereby exhibiting flexibility and competitiveness. This is particularly noteworthy as sequencing costs continue to decline. It would be valuable if the authors could consider including a comparison of their method with, for instance, CAPP-Seq in their analysis.

Response: Thank you for your advice. CAPP-Seq is a promising sequencing technology that can improve detection sensitivity through deep sequencing. Therefore, In the revised introduction section, we added the information of “Sequencing-based methods, such as next-generation sequencing, can provide information for multiple genes, while it poses difficulty in detecting allelic mutants present below 0.1% with routine sequencing depth. Cancer personalized profiling by deep sequencing (CAPP-seq) is a highly sensitive NGS-based technology developed at Stanford University for ctDNA detection. It's gradually being applied in various cancer studies for diagnosis, profiling, and treatment response tracking. However, PCR-related methods may be more economical and faster for detecting several target genes”, (in line 66) marked in yellow.

In our study, the *EGFR* assay kit, based on the Super-ARMS method, has been certified by the CFDA and is considered the gold standard for EGFR mutation testing. Therefore, we used the *EGFR* assay kit as a comparison, referring to some relevant literature^{1,2}. In addition, to further validate the sensitivity of our method, we conducted a comparison of EGFR mutation detection using the well-recognized high-sensitivity technique, ddPCR. We think it is more adequate and comprehensive to compare HiCASE with both methods simultaneously in this study. CAPP-

Seq is a multi-target gene mutation detection method that also requires high sample quality. Thank you again for your valuable suggestion. We are actively developing a multi-target high-sensitivity technology based on the HiCASE method. For the subsequent minimal residual disease (MRD) detection, we will prioritize using CAPP-Seq as a comparison.

Reference

1. Jiang, H. et al. Validation of a highly sensitive Sanger sequencing in detecting EGFR mutations from circulating tumor DNA in patients with lung cancers. *Clinica Chimica Acta* 536, 98-103 (2022).
2. Xu, J. et al. A large-scale, multicentered trial evaluating the sensitivity and specificity of digital PCR versus ARMS-PCR for detecting ctDNA-based EGFR p. T790M in non-small-cell lung cancer patients. *Translational Lung Cancer Research* **10**, 3888 (2021).

Comment 2: I appreciate the authors' attention to the input DNA requirement, particularly as HiCASE necessitates less input. It would be insightful to determine the minimum required DNA amount (in addition to plasma volume).

Response: Thank you for your advice. Due to variations in mutation frequencies among plasma samples, the minimum detectable cfDNA amount also varies. Therefore, we determined the minimum cfDNA amount using *EGFR* cfDNA standards. In our method, detecting 0.1% VAF for the L858R or 19dels mutation required 5 ng of cfDNA input (**Fig. S7a-b**). We also analyzed the minimum cfDNA input required for detecting EGFR mutations in plasma samples. The result was added in **Supplementary Figure 8**. As shown in **Fig. S8**, the minimum cfDNA input was derived from **Figure 5a**. For the detection of the L858R mutation, it amounted to 0.14 ng in sample 6, with a VAF of 10.8%. In the case of detecting the 19dels mutation, it was 0.21 ng in sample 18, demonstrating a VAF of 14.2%. In the revised manuscript, the content: “We also evaluated the minimum cfDNA input from plasmas in samples in the 18 positive cases. For detecting the L858R mutation, it amounted to 0.14 ng in sample 6, with a VAF of 10.8%. As the 19dels mutation, this input was 0.21 ng in sample 18, corresponding to a VAF of 14.2%” (in line 217) was added in the result section, marked in yellow.

Supplementary Figure 7. Comparison of different amounts of cfDNA used by the HiCAGE and ddPCR assays. a-b The results of detecting L858R and 19dels mutations using the HiCAGE assay with different amounts of cfDNA standards in 0.1% VAFs. Error bars represent the mean \pm S.D., $n = 3$ technical replicates. **c** The detection of L858R and 19dels used by ddPCR in different amounts of cfDNA standards with 0.1% VAFs.

Supplementary Figure 8. The minimum amount of cfDNA for detecting L858R and 19dels mutation calculated from Figure 5a.

Figure 5. Plasma volume needed for detecting *EGFR* mutations with HiCAGE and ddPCR. a Detecting 18 plasma samples including L858R mutation (N=9) and 19dels (N=9) for determining the plasma volume of HiCAGE assay. **b** The 18 plasma samples deriving from **a** were also detected by ddPCR assay with 40 μL, 20 μL, 10 μL and 5 μL. $n = 3$ technical replicates.

Comment 3: Developing a specific crRNA for each new mutation is a crucial aspect of the assay. The authors address this challenge by designing a range of crRNA (up to 8) for each site and testing their specificity and sensitivity. Do the authors consider this as a significant drawback of the assay, given that the design and testing process is not as straightforward as, for example, primers for PCR or the design for CAPP-Seq sites?

Response: The Cas13a-related crRNA can be designed through software or manual methods, which is a relatively mature process. In this study, we avoided the tedious preparation of crRNA and instead directly added the transcription template of crRNA (DNA) into the reaction system, thereby enhancing the efficiency of screening optimal crRNA. The PCR primers of specifically

amplifying mutant type also present some challenges to design because the primers may also amplify the wild type when the mutation frequency is low, as seen in the Super-ARMS method. This could result in false positives and reduced detection sensitivity. Obviously, the CAPP-Seq method offers significant advantages in the high-throughput detection of large-scale genes, but its operation is relatively complicated compared with PCR-based methods for detecting several target genes.

Comment 4: Regarding Fig. 2 h-i: I suggest modifying the legends of the heatmap to avoid displaying a spectrum between 0 and 1, as only 0 and 1 can be called. Adding explanations for what 0 and 1 represent in the legend would enhance readability.

Response: Thank you for your advice. In the revised manuscript, we substituted “N” and “P” for “0” and “1”

Comment 5: In Fig. 4 c-e, the legends mention $n=3$, but the actual figures (e.g., Fig. 4C) indicate different numbers. Shouldn't the n value for Fig. 4d-e be much higher, considering the inclusion of 120 NSCLC patients and 20 healthy donors?

Response: Thank you for your reminder. In **Fig. 4c**, the n value indicates different numbers, but in **Fig. 4 d-e**, the n value indicates the technical replicates. To avoid confusion arising from the two representations, we opted to change “ $n=3$ ” to “ $n=3$ technical replicates” and replaced “ n ” with “N” to indicate sample numbers.

Reviewer 3 (Major or Minor Revisions)

Comment 1: Value of restriction enzyme digestion: The key advance of the HiCASE method presented here over other CAS based methods like SHERLOCK is the combination of CRISPR/Cas13a based mutation detection with selective digestion of WT fragments through restriction enzymes (RE). However, the authors fail to convincingly show that the incorporation of RE improves detection. While Fig 2C does show an advantage for L858R detection when using RE digestion, I find it remarkable that the overall fluorescence signals for mutant containing conditions go up and only the WT-only signal goes down. The authors state that the

RE step is supposed to reduce the background, but it seems to selectively do so in the condition with only WT fragments. Do the authors have an explanation for that? The authors should show similar experiments for other mutations, and also demonstrate the advantage of the RE step in primary patient samples.

Response: Thank you for your reminder. We think that without RE digestion, the numerous wild fragments in the reaction system interfere with the detection efficiency of Cas13a/crRNA for mutant fragments, leading to decreased fluorescence values and lower sensitivity. Conversely, when employing RE digestion to eliminate almost all wild fragments, followed by purification using the kit, it will enhance efficiency in identifying mutant fragments by Cas13a/crRNA. In the revised Results section, we added the statement: “Furthermore, digesting wild fragments with RE can reduce their interference in fluorescence detection, resulting in a noticeable improvement in fluorescent values for detecting mutant fragments,” (in line 129) marked in yellow. In addition, the overall fluorescence value for the 19dels mutation notably improved following RE digestion, as depicted in **Fig. S4**. We added the result in the supplementary materials.

Supplementary Figure 4. The detection of different VAFs of *EGFR* 19del plasmids using two approaches. The data was analyzed using two-tailed Student’s t test. $n = 3$ technical replicates, error bars represent the mean \pm S.D; *** $P < 0.001$, and **** $P < 0.0001$; ns, not significant.

Comment 2: There seems to be a huge discrepancy between the synthetic DNA testing experiments, and the experiments around patient plasma samples. For the patient plasma

samples only 40 μ l input plasma were used. Why do the authors not refer to masses as in the remainder of the manuscript? With plasma cfDNA concentrations usually being around 10ng/mL, 40 μ l of plasma would only account for 0.4ng DNA reflective of 100-200 tumor genomes as input. In contrast for synthetic DNA 20-40 ng was used, which seems vastly different. The authors should report the input DNA mass used for each of the 120 patient samples.

Response: Indeed, while the concentration of cfDNA in the blood is relatively low, there is substantial variation among cancer patients. Some literature revealed a range spanning from 0 to >1,000 ng/mL of blood, with an average cfDNA concentration of 180 ng/mL¹⁻³. Phallen et al. analyzed plasma samples and found that the cfDNA concentrations were between 1 and 1000 ng/mL⁴. In our study, we confirmed that about 5 ng of cfDNA standards could detect EGFR L858R and 19dels with a VAF of 0.1%. However, for lower VAFs such as 0.02%, a higher amount of cfDNA is required. We utilized plasma volume as a quantifiable indicator, which serves as an assessment for blood collection. We also quantified the concentrations of cfDNA extracted from samples and the result was added in **Supplementary Figure 6a**. The concentrations of cfDNA extracted from lung cancer plasmas ranged from 9 ng/mL to 612 ng/mL, with a median of 62 ng/mL. In the revised manuscript, the content: The concentrations of cfDNA extracted from NSCLC plasmas ranged from 9 ng/mL to 612 ng/mL, with a median concentration of 62 ng/mL (Fig. S6a),” (in line 187) was added in Results section marked in yellow. Additionally, in five cases where EGFR mutations were verified positive in tissue, they were not detected in cfDNA, likely due to very low levels falling below the detection range.

Supplementary Figure 6. cfDNA in healthy individuals and patients with lung cancer. a The concentration of cfDNA in the plasma of healthy individuals and lung cancer patients. **b** The mutant allele fraction of EGFR detected by ddPCR in healthy individuals and lung cancer patients. Medians for each group are represented by the black bars.

Reference

1. Schwarzenbach, H., Stoeckl, J., Pantel, K. & Goekkurt, E. Detection and monitoring of cell-free DNA in blood of patients with colorectal cancer. *Ann. N. Y. Acad. Sci.* 1137, 190–196 (2008).
2. Chun, F. K. et al. Circulating tumour-associated plasma DNA represents an independent and informative predictor of prostate cancer. *BJU Int.* 98, 544–548 (2006).
3. Schwarzenbach, H., Hoon, D. S. & Pantel, K. Cell-free nucleic acids as biomarkers in cancer patients. *Nat Rev Cancer* **11**, 426–437 (2011).
4. Phallen, J. et al. Direct detection of early-stage cancers using circulating tumor DNA. *Science translational medicine* **9**, eaan2415 (2017).

Comment 3: What was the mean AF in each of the 120 patient samples. In order to detect mutations from just 100-200 genomes (see above), the ctDNA concentration must be really high. Orthogonal data by conventional sequencing must be provided in order to make the data interpretable.

Response: Thank you for your advice. In this study, we focused on *EGFR* mutations as the target and evaluated the sensitivity of HiCASE, comparing it with ddPCR and Super-ARMS methods. Therefore, we employed ddPCR for absolute quantification of the *EGFR* gene and calculated the VAFs. The VAFs of L858R and 19dels were added in **Supplementary Figure 6b**. The 23 samples were identified with L858R or 19dels mutations by ddPCR, exhibiting VAFs ranging from 0.13% to 15.4%, with a median of 1.5%. The content: “Among 23 samples identified with L858R or 19dels mutations via ddPCR, the VAFs ranged from 0.13% to 15.4% (Fig. S6b), indicating HiCASE can detect EGFR mutations in plasma samples with a VAF as low as 0.13%,” was added in the Results section marked in yellow.

Supplementary Figure 6. cfDNA in healthy individuals and patients with lung cancer. a The concentration of cfDNA in the plasma of healthy individuals and lung cancer patients. **b** The mutant allele fraction of EGFR detected by ddPCR in healthy individuals and lung cancer patients. Medians for each group are represented by the black bars.

Comment 4: I also did not really understand what kind of synthetic DNA was used for the technical experiments. The authors partially refer to plasmids (circular?) or cfDNA standards. The authors should clarify on fragment length and whether the synthetic DNA (WT and mutant) only covered the EGFR locus or the full genome. This is particularly important since the dilutions were apparently made by mass, and the mass of a full genome is \gg larger than the mass of a copy of EGFR.

Response: Target DNA fragments were synthesized to screen optimal crRNA and evaluate RE cleavage efficiency. Each EGFR mutation subtype comprises wild-type and mutant variant. Synthetic DNA of the wild type or mutant type was about 400 bp and inserted into the PUC57 plasmid, respectively. To establish a standard system for testing clinical samples, we purchased EGFR cfDNA standards, which are certified by the authoritative institution and have been used in many literatures^{1,2}. In theory, the cfDNA standards are used to mimic the cfDNA extracted from blood samples. It is obtained by fragmenting the genome, followed by the absolute quantification of the target fragments. Therefore, in this study, we used EGFR cfDNA standards to evaluate the sensitivity and specificity for L858R, 19dels and T790M before testing clinical samples.

Reference

1. Liu, Q. *et al.* Argonaute integrated single-tube PCR system enables supersensitive detection of rare mutations. *Nucleic Acids Res* **49**, e75 (2021).
2. Gootenberg, J. S. *et al.* Nucleic acid detection with CRISPR-Cas13a/C2c2. *Science* **356**, 438-442 (2017).

Comment 5: Description of methods is very sparse especially given the authors aim to introduce a new method here. The authors need to add details on their precise workflow when starting from synthetic DNA as well as patient samples. Was the same number of PCR cycles done for technical experiments and for the patient samples? Was synthetic DNA amplified at all? Given the high number of PCR cycles reported in the methods section (36 cycles!) I would expect to see a significant background through the introduction of error through PCR errors which would likely preclude sensitive detection at levels of 0.01%.

Response: Thank you for your advice. Synthetic DNA was employed to evaluate the sensitivity of HiCASE due to its ease of acquisition and quantification. To detect cfDNA from clinical blood samples, we established the standard detection system of this method. Different VAFs of cfDNA standards were used to determine the sensitivity and specificity of HiCASE and establish the cutoff value. We conducted ten repetitions of wild type cfDNA detection using HiCASE, with no observable fluorescent signal detected. It indicated that the potential errors induced by PCR cycles are minimal and insufficient to disturb the detection efficiency of 0.01%. We added some details in the methods part to make the HiCASE workflow clearer. The revised content was marked in yellow as follows:

To evaluate the sensitivity of HiCASE detection system, mutant-type, and wild-type plasmids were mixed at different VAFs, including 10, 1, 0.1, and 0.01%. The wild-type plasmid was maintained at 10^{-3} ng/ μ L, while the mutation plasmids were gradually added in the range from 10^{-4} to 10^{-8} ng/ μ L. Different VAFs of plasmids were amplified by PCR with a 30 μ L reaction system containing 1 \times Taq mix, 0.5 μ M of primers, 1 μ L of plasmids, and DEPC H₂O. The PCR amplification condition consisted of pre-incubation at 95°C for 5 min; 36 cycles of 95°C for 30 s, 56°C for 15 s, and 72°C for 30 s; and holding at

72°C for 5 min. About 150 ng of PCR products were digested by RE at 37°C for 40 min. Then, the digested products were purified using the Select-a-Size DNA Clean & Concentrator Kit. Finally, they were detected by Cas13a/crRNA reaction for 40 min, The reaction consisted of 25 mM Tris HCl, 9 mM MgCl₂, 100 ng Cas13a, 5 nM crRNA transcript template, 10 mM rNTP, 15 U T7 RNA polymerase, 8 U RNA inhibitor, 0.25 μM FAM-labeled ssRNA reporters, digested products, 50 ng total human RNA, and DEPC H₂O added to the total mixture volume of 20 μL. The fluorescent signals were collected with Real-Time PCR Systems.

To establish a standardized HiCASE detection system for clinical sample analysis, To further verify the clinical utility of the HiCASE assay, cfDNA Reference Standards (Horizon Discovery, Cambridge, UK, Cat# HD825) were purchased from a commercial supplier. cfDNA standards consisted of *EGFR* del15, L858R, and T790M with different VAFs (0.1, 1, and 5%) at a concentration of 20 ng/μL. To achieve VAFs of 0.2 and 0.02%, we mixed 16 ng of wild-type DNA with 4 ng of mutant-type cfDNA at 1% and 16 ng of wild-type DNA with 4 ng of mutant-type cfDNA at 0.1%. Different VAFs (0.1, 0.2, 1, 0.2, and 5%) cfDNA standards were amplified with a 30 μL reaction system including 15 μL of 2×Taq mix, 0.5 μM of primers, 1 μL of cfDNA standard, and DEPC H₂O. The PCR cycling program started with a pre-incubation at 95°C for 5 min; 36 cycles of 95°C for 30 s, 56°C for 15 s, and 72°C for 30 s; and holding at 72°C for 5 min. About 150 ng of PCR products were digested by RE at 37°C for 40 min. The reaction system consists of 5 μL of 10× Cutsmart buffer, 10 U RE, 150 ng DNA, and adding DEPC H₂O to a final volume of 50 μL. 20 ng cfDNA. A total of 150 ng of PCR products was digested by RE at 37°C for 40 min. Following this, the digested products were purified, added to the Cas13a/crRNA reaction, and incubated for 40 min, The reaction consisted of 25 mM Tris HCl, 9 mM MgCl₂, 100 ng Cas13a, 5 nM crRNA transcript template, 10 mM rNTP, 15 U T7 RNA polymerase, 8 U RNA inhibitor, 0.25 μM FAM-labeled ssRNA reporters, digested products, 50 ng total human RNA, and DEPC H₂O added to the total mixture volume of 20 μL. The FAM fluorescent signal was acquired using the Real-Time PCR Systems.

For identifying *EGFR* mutations in plasma samples, 2 μL of extracted cfDNA from plasma was detected by HiCASE. The operational process was consistent with the standardized HiCASE detection system. The detection process for all clinical samples follows this protocol.

Besides, the cfDNA extracted from 40, 20, 10, 5, and 2 μL of plasma was employed for detection using the HiCASE assay.

Comment 6: Manuscript presentation: The manuscript is partially hard to follow. For instance, the visualization of introduced RE sites needs to be improved. In the current depiction, it is unclear how the mutant primers +/- EGFR mutations lead to a specific RE site.

Response: Thank you for your advice. We added a schematic diagram of PCR site-directed mutagenesis in **Supplementary Figure 2**. The principle of site-directed mutagenesis is that Taq DNA polymerase lacks 3' to 5' exonuclease activity, so when there is a mismatch between the primer and template, Taq DNA polymerase cannot recognize it. It will continue amplification using the primer sequence, allowing the newly generated DNA fragment contains the mutated bases. The content: "Due to Taq DNA polymerase lacking 3' to 5' exonuclease activity, it cannot identify a mismatch between the primer and the template. After several cycles of PCR, most of the products will harbor the mismatch base (Fig. S2)," (in line 140) was added in the Result section marked in yellow. In addition, we added a diagram in **Fig. 6a** to illustrate the generation and elimination of the C797S/T790M RE sites, making it clearer.

Supplementary Figure 2. Schematic diagram of PCR site-directed mutagenesis. The purple point

indicates the original base and the red point indicates the mutated base. The primer carrying the mutation is employed to replace the original base via PCR amplification. Since Taq DNA polymerase lacks 3' to 5' exonuclease activity, it cannot identify a mismatch between the primer and the template. After the initial cycle, the product yielded ssDNA containing the mutated base. After the second cycle, the product generated dsDNA with the mutated base. By the third cycle, it consisted of 50% dsDNA with the mutated base, and after four cycles of amplification, nearly all the product contained mutated dsDNA.

Figure 6. Detecting T790M and C797S mutations in different positions using the HiCASE assay. a Schematic diagram of restriction enzyme sites formed after PCR amplification of wild-type, T790M, C797S, and T790M-cis-C797S mutation. Red indicated the mutated base of wild type and yellow for RE site. **b** The result of Sanger sequencing for EGFR C797S mutation after site-directed mutagenesis. Yellow indicated the base to be mutated and red for mutated base of wild type. **c** The electropherogram of PCR products of T790M/C797S before and after digesting by the Hpych4V enzyme. (1) T790M/C797S wild type; (2) T790M mutation; (3) C797S mutation; (4) T790M-cis-C797S mutation; **d** The purified products of T790M/C797S with cis/trans fragments after digestion. (1) T790M-cis-C797S fragments; (2) T790M-

trans-C797S fragments. **e** The detection of T790M and C797S cis/trans mutation with HiCAGE assay. n=3 n = 3 technical replicates, two-tailed Student's t test, error bars represent the mean \pm S.D; *P <0.05, **P <0.01, ***P < 0.001, and ****P < 0.0001.

Comment 7: Fig 4c is not interpretable since ground truth information is not provided. Should be replaced through confusion matrices.

Response: Thank you for your advice. We used the confusion matrix to replace **Fig 4c** for showing the detected result of the T790M mutation.

Figure 4. Detecting *EGFR* mutations of blood samples using HiCAGE, ddPCR, and Super-ARMS assays. **a** The flowchart for detecting *EGFR* mutations of blood samples using HiCAGE, ddPCR and Super-ARMS assays. 120 blood samples from 85 adenocarcinoma and 35 squamous cell carcinomas were collected to determine the detection sensitivity and specificity of three assays. 20 blood samples from health donors were involved to verify the specificity of the HiCAGE assay. **b** The detection of

L858R, 19dels and T790M mutations using the HiCASE assay. **c** The confusion table of detecting result of T790M mutation in different methods. The detection of tissue samples was used as the gold standard. **d-e** Comparison the sensitivity and specificity of HiCASE, ddPCR and Super-ARMS assays for detecting L858R and 19dels mutations. $n = 3$ technical replicates. Error bars represent the mean \pm S.D.

Comment 8: Supp. Fig 4. Every input mass needs a WT control since it is expected to impact background.

Response: Thank you for your advice. In this experiment, we added a WT control for each input mass. The fluorescence values of different input WT controls did not vary significantly, as the input amount of PCR product for RE digestion is standardized.

Supplementary Figure 7. Comparison of different amounts of cfDNA used by the HiCASE and ddPCR assays. a-b The results of detecting L858R and 19dels mutations using the HiCASE assay with different amounts of cfDNA standards in 0.1% VAFs. Error bars represent the mean \pm S.D., $n=3$. **c** The detection of L858R and 19dels used by ddPCR in different amounts of cfDNA standards with 0.1% VAFs.

REVIEWERS' COMMENTS:

Reviewer #2 (Remarks to the Author):

The authors responded to all of my concerns and addressed them in a reasonable manner.

Reviewer #3 (Remarks to the Author):

The authors have carefully addressed the raised concerns. I have no further comments.